# SEMI-SUPERVISED BOOSTING VIA SELF LABELLING

## ABSTRACT

Attention to semi-supervised learning grows in machine learning as the price to expertly label data increases. Like most previous works in the area, we focus on improving an algorithm's ability to discover the inherent property of the entire dataset from a few expertly labelled samples. In this paper we introduce Boosting via Self Labelling (BSL), a solution to semi-supervised boosting when there is only limited access to labelled instances. Our goal is to learn a classifier that is trained on a data set that is generated by combining the generalization of different algorithms which have been trained with a limited amount of supervised training samples. Our method builds upon a combination of several different components. First, an inference aided ensemble algorithm developed on a set of weak classifiers will offer the initial noisy labels. Second, an agreement based estimation approach will return the average error rates of the noisy labels. Third and finally, a noise-resistant boosting algorithm will train over the noisy labels and their error rates to describe the underlying structure as closely as possible. We provide both analytical justifications and experimental results to back the performance of our model. Based on several benchmark datasets, our results demonstrate that BSL is able to outperform state-of-the-art semi-supervised methods consistently, achieving over 90% test accuracy with only 10% of the data being labelled.

## 1 INTRODUCTION

The rise of the Internet has made it easy to collect massive amounts of data to perform machine learning tasks. However, providing quality labels to each of the samples collected within these large datasets is a long and expensive process. There is a rich literature aiming to alleviate this issue, including using techniques from unsupervised machine learning and crowdsourcing. In this paper, we approach the problem by combining concepts from crowdsourcing, learning with noisy data and boosting to propose a novel framework: Boosting via Self Labelling (BSL).

Our aim is to develop and leverage i) machine learning and estimation approaches to create self-labels for unlabelled instances, and ii) a noise-resistant learning procedure to speed up the performance of the seminal AdaBoost algorithm. The framework consists of mainly three steps:

1. Accurately predict labels for the unlabelled part of the data using a set of supervised classifiers trained upon a small labelled dataset. Each classifier in our set is analogous to an agent in a crowdsourcing setting. As a result, an inference method can be used to aggregate each of the classifiers predictions and output an accurate noisy label for each of the unlabelled data points.

2. The second step aims to estimate the noise rate of the generated noisy labels by checking how often the noisy labels generated in step 1 would agree with a particular classifier. This second order statistic suffices to return us the error rates.

3. The third step of our approach looks at producing a robust boosting method that is trained over the generated noisy data. Classically AdaBoost does not perform well under noisy data compounding errors for each point and progressively creating a worse classifier. The third step introduces a noise resistant version of AdaBoost which relies on the noise and error rate estimated in step 2. This results in a final classifier which can be compared against different semi-supervised algorithms.

BSL builds upon mainly two lines of similar works on boosting without cleanly labelled data:

1. *Semi-supervised boosting* (Fujino et al., 2005; Blum & Mitchell, 1998; Laine & Aila, 2016; Grandvalet & Bengio, 2005) uses semi-supervised algorithms (e.g., clustering) to generate ar-

tificial or proxy labels and boost accordingly (which we compare with). Existing methods that apply self-generated labels directly to boosting will fail as the noises in the labels will accumulate while boosting, especially when the noise rates are high - this is what we observed in our experiment results. Our idea is a couple of steps further: we introduce a bias correction procedure into boosting, and explicitly estimate the noises in generated labels. We also introduced an inference framework for generating these labels at first place.

2. *Noise resistant boosting* (Bootkrajang & Kabán, 2013) addresses boosting algorithms which are susceptible to noisy data and proposing variants which can perform under a certain noisy conditions. A set of noisy labels, as well as the knowledge of the noises, are often assumed to be known. We do not require neither - we will self-generate the labels for unlabelled instances and learn their error rates.

Our contributions summarize as follows:

1. We propose a novel self-labelling boosting algorithm (BSL) which is able to outperform present state-of-the-art semi-supervised algorithms.

2. As two key components of our self-labelling framework, we contribute i) a new formulation of a noise resistant Adaboost algorithm which corrects the noises in the labels - this is important in a boosting process, because otherwise the label noises will accurate while boosting; ii) a label error estimation procedure without accessing the ground truth labels.

3. We offer both theoretical guarantees, as well as experimental evidence to the performance of our framework. We conducted an extensive set of experiments to verify the effectiveness of BSL. In the different datasets we ran our algorithm on, our method consistently outperforms many other algorithms by more than **20%**. In the cancer dataset when 10% of the data was labelled, the second best algorithm (Semiboost) under performed our algorithm by **66%** (relative performance). Theoretically we are able to show the convergence of our noise-resistant AdaBoost subroutine under symmetric error rate assumption.

The rest of the paper organizes as follows. We survey the most relevant work in the rest of the section. Preliminaries are introduced in Section 2. Section 3 presents a noise-resistant AdaBoost algorithm. Our Boosting via Self Labelling framework is introduced in Section 4. Section 5 presents our experiment results. Section 6 concludes our paper. All missing details can be found in Appendix.

## 1.1 RELATED WORKS

Our work has been inspired by three different lines of research:

*Semi-Supervised Learning*: Research in this avenue looked at creating accurate labels in the presence of limited labelled data. Work started with some basic algorithms (Cortes & Vapnik, 1995; Fujino et al., 2005; Demiriz et al., 1999; Blum & Mitchell, 1998), but escalated to complex systems (Lee; Miyato et al., 2018; Laine & Aila, 2016; Grandvalet & Bengio, 2005). A good survey in this area was done by Zhu (Zhu, 2005).

*Crowdsourcing*: Inference crowdsourcing methods have played a huge part in uncovering true labels from multiple noisy labels. Work in the area has ranged from EM algorithms (Dawid & Skene, 1979; Raykar et al., 2010; Smyth et al., 1995; Karger et al., 2011) to variational inference methods (Liu et al., 2012; Karger et al., 2014; Whitehill et al., 2009; Welinder et al., 2010; Raykar & Yu, 2012). (Chon et al., 2012) wrote a good survey for this topic. In an ensemble setting, crowdsourcing has appeared in numerous works include, most famously, (Kim & Ghahramani, 2012). However, many of these results do not consider the noise present in the final aggregated value, which can lead to accumulated errors in a boosting setting.

*Boosting*: Starting with the work (Freund & Schapire, 1999), research in AdaBoost has expanded to all areas of machine learning. Some of the recent work in the area has looked into improving the performance and pitfalls the original algorithm faced (Rätsch et al., 2001; Hastie et al., 2009; Domingo et al., 2000; Schapire, 2013; Bootkrajang & Kabán, 2013).

Our work builds upon previous work done in these areas, using similar ideas to construct a unique method. One notable piece of work is Semiboost (Zhu, 2005). This algorithm takes a similar approach, introducing a boosting framework that improves upon existing classifiers to provide a good

classifier in the semi-labelled setting. However, Semiboost uses an unsupervised learning approach using a similarity matrix based on labelled points and unlabelled points. Semiboost requires a similarity equation to run, and this can be hard to achieve in practice. Pseudo-labelling is one of many neural network approaches to the limited labelled dataset problem. Pseudo-labelling assigns labels to the unlabelled data and then trains a neural network on the combination of the clean and noisy labels. However, Pseudo-labelling and algorithms like it, face many different requirements. Algorithms in this class require a sufficient amount of supervised data to work optimally. They also need the classes to be clustered within the data, and the labelled data to not adhere to the same distribution as the unlabelled data (Oliver et al., 2018). Learning in noisy data has also been a research focus that runs parallel to our work. In (Natarajan et al., 2013), Natarajan et al. propose a noisy learning algorithm which performs significantly better than other noise resistant algorithms. However, the method requires knowledge on the amount of noise inside of the dataset before training and in a limited label dataset, which is hard to get in practice. In contrast to these related works, the algorithm we introduce in this paper does not need prior knowledge about the dataset for it to perform and the number of labelled points does not adversely affect the accuracy of the classifier generated.

## 2 PRELIMINARY AND PROBLEM FORMULATION

Assume that $\mathcal{D}$ is the underlying true distribution generating $n$ *iid* examples $(x_i, y_i)_{i=1}^n$, where each example consists of a *feature vector* $x_i = [x_{i,1}, x_{i,2}, \ldots, x_{i,d}] \in X \subseteq \mathbb{R}^d$, and a *label* $y_i \in \{-1, +1\}$ ($\sim Y$). We assume the true labels of $\tau$ examples is available where $\tau < n$ such that $(x_i, y_i)_{i=1}^\tau$, denoted as $\{(x_1, y_1), (x_2, y_2), \cdots, (x_{|N_L|}, y_{|N_L|})\} := \mathcal{N}_L$ ; while the rest of $n - \tau$ samples $(x_i)_{i=\tau}^n$ is unlabelled, $\{x_1, x_2, \cdots, x_{|N_U|}\} := \mathcal{N}_U$. Let $N_U$ and $N_L$ be the set of indices that make up $\mathcal{N}_U$ and $\mathcal{N}_L$ respectively.

Assume that after an initial classifier $f : X \to \{-1, +1\}$ is trained on $\mathcal{N}_L$ and applied on the unlabelled dataset such that $f(x_i) \to \tilde{y}_i, x_i \in \mathcal{N}_U$. Then $\mathcal{N}_{U_{noisy}}$ denotes the noisy data set produced $(x_i, \tilde{y}_i)_{i=1}^{|N_U|}$. Let $\mathcal{N}_{noisy}$ be the combination of $\mathcal{N}_L$ and $\mathcal{N}_{U_{noisy}}$ such that for $i = 1$ to $|N_{noisy}|$: $\{(x_i, \tilde{y}_i)$, if $(x_i, \tilde{y}_i) \in \mathcal{N}_{U_{noisy}}$; $(x_i, y_i)$, if $(x_i, y_i) \in \mathcal{N}_L\}$. Let $N_{noisy}$ be the set of all indices within $\mathcal{N}_{noisy}$. Assume that $\mathcal{N}_{noisy}$ follows as class-conditional random noise model such that: $\forall n = 1, 2, ..., |N_{noisy}|$: $\mathbb{P}(\tilde{y}_i = -1|y_i = +1, x_n) = \rho_+$, $\mathbb{P}(\tilde{y}_i = +1|y_i = -1, x_n) = \rho_-$ and $\rho_+ + \rho_- < 1$. Our goal is to learn a classifier $f : X \to \{-1, +1\}$ trained on $\mathcal{N}_{noisy}$ that minimizes the risk of $f$ *w.r.t* to the 0-1 loss function $R_{\mathcal{D}}(f) = \mathbb{E}_{(x_i, y_i) \sim \mathcal{D}}[\mathbb{1}(f(x_i) \neq y_i)]$.

### 2.1 OUR PROBLEM: BOOSTING VIA SELF LABELLING

We introduce the setting for AdaBoost (Freund & Schapire, 1999). The key idea is, at step $t$,

- Maintain a weight $D_i(t)$ for each data instance $(x_i, y_i) \in \mathcal{N}_L$.
- Train a weak learner $f_t$ according to the weighted data distribution.
- The final hypothesis $F$ is a linear combination of each $f_t$ trained at every step $t$.

Let $\mathcal{N}_{miss}$ be the set of all $(x_i, y_i) \in \mathcal{N}_L$, such that $f_t(x_i) \neq y_i$. The goal is to increase the weight of mis-classified points $D_i(t), (x_i, y_i) \in \mathcal{N}_{miss}$ to encourage classifier $f_{t+1}(\cdot)$ to focus on correctly classifying $\mathcal{N}_{miss}$:

$$D_t(i+1) = \frac{D_t(i) \cdot \exp(-\alpha_t \cdot f_t(x_i) \cdot y_i)}{Z_t},$$

where $Z_t = \sum_{i \in N_L} D_t(i) \cdot \exp(-\alpha_t \cdot f_t(x_i) \cdot y_i)$ is a normalization factor. AdaBoost creates a final hypothesis in the additive form: $F(x_i) = \sum_{t=1}^T \alpha_t f_t(x_i)$, where $x_i$ is a test sample.

Our goal, and a short coming with AdaBoost, is classifying a dataset where some of the $y_i$ in $(x_i, y_i) \in \mathcal{N}_{noisy}$ are noisy. Because AdaBoost uses a exponential loss function it is inherently susceptible to noisy labels. We propose a new loss function that removes bias:

$$D_{t+1}(i) = \frac{D_t(i) \cdot \exp(-\alpha_t \cdot \tilde{\ell}(x_i, \tilde{y}_i))}{Z_t}, \ \forall i \in N_{noisy}$$

Our goal is to define a function $\tilde{\ell}(\cdot)$ that can help us evaluate an unlabelled instance. This function $\tilde{\ell}(\cdot)$ will allow us to adjust to noisy labels within $\mathcal{N}_{noisy}$. Our algorithm runs in two main stages. It

first applies noisy labels for unlabelled instances in our dataset, and then creates a final hypothesis $F$ using a noise-resistant variant of AdaBoost where we define $\tilde{\ell}(\cdot)$ .

## 3 NOISE-RESISTANT ADABOOST

We first extend a learning with noisy data approach to the boosting setting, following the work in (Natarajan et al., 2013). Suppose the examples have the following homogeneous error rates:

$$\rho_+ := \mathbb{P}_{x_i}(\tilde{y}_i = -1 | y_i = +1), \rho_- := \mathbb{P}_{x_i}(\tilde{y}_i = +1 | y_i = -1)$$

Suppose we know these error rates in this section. Boosting over the above noisy examples will lead to a biased training process when the label noises are sufficiently large. Our approach is a straight-forward adaptation from a noise correction mechanism adopted in supervised learning (Natarajan et al., 2013): defining surrogate loss function $\tilde{\ell}$ on noisy labels (for an arbitrary loss function $\ell$)

$$\tilde{\ell}(f(x_i), \tilde{y}_i = +1) := \frac{(1 - \rho_-)\ell(f(x_i), +1) - \rho_+\ell(f(x_i), -1)}{1 - \rho_+ - \rho_-}, \tag{1}$$

$$\tilde{\ell}(f(x_i), \tilde{y}_i = -1) := \frac{(1 - \rho_+)\ell(f(x_i), -1) - \rho_-\ell(f(x_i), +1)}{1 - \rho_+ - \rho_-}. \tag{2}$$

A nice property of above estimator is its unbiasedness (Natarajan et al., 2013): $\mathbb{E}_{\tilde{y}_i | y_i}[\tilde{\ell}(f(x_i), \tilde{y}_i)] = \ell(f(x_i), y_i)$. We adapt this idea to AdaBoost. Replace $\ell(\cdot)$ with the following loss measure as adopted in AdaBoost: $\ell(f(x_i), y_i) = f_t(x_i) \cdot y_i$. Define $\omega_+ := \frac{1 - \rho_- + \rho_+}{1 - \rho_- - \rho_+}, \omega_- := \frac{1 - \rho_+ + \rho_-}{1 - \rho_- - \rho_+}$ and

$$\hat{\epsilon}_t^+ := \mathbb{P}_{x | \tilde{y} = +1}(f_t(x) \neq \tilde{y}), \ \hat{\epsilon}_t^- := \mathbb{P}_{x | \tilde{y} = -1}(f_t(x) \neq \tilde{y}), \tag{3}$$

and $\hat{\epsilon}_t := \max\{\hat{\epsilon}_t^+, \hat{\epsilon}_t^-\}$ and the following $\alpha_t$s

$$\alpha_t^+ = \frac{1}{2\omega_+} \ln \frac{1 - \hat{\epsilon}_t}{\hat{\epsilon}_t}, \ \ \alpha_t^- = \frac{1}{2\omega_-} \ln \frac{1 - \hat{\epsilon}_t}{\hat{\epsilon}_t} \tag{4}$$

Then update $D_i(t)$ as follows:

$$D_i(t + 1) := \frac{D_i(t) \cdot \exp\left(-\alpha_t^{sign(\tilde{y}_i)} \tilde{\ell}(f(x_i), \tilde{y}_i)\right)}{Z_t} \tag{5}$$

where $Z_t$ is again the normalization factor. The reason that we need to define two learning rate $\alpha_t$s is because the losses are weighted differently for the noisy labels. and $\omega := \max\{\omega_+, \omega_-\}$, $\gamma_t := \frac{1}{2} - \hat{\epsilon}_t$, and $\epsilon(\delta, n, \rho_+, \rho_-) := \omega \cdot \sqrt{\frac{n \ln \frac{2}{\delta}}{2}}$. Since we have defined two $\alpha_t$s on the training data based on the noisy labels, we define $\alpha_t := \frac{\alpha_t^+ + \alpha_t^-}{2}$, and let the final output classifier be $F(x) = sign\left(\frac{\sum_{t=1}^{T} \alpha_t f_t(x)}{T} > 0\right)$. We prove the following performance guarantee when $\rho_+ = \rho_-$:

**Theorem 1** *With probability at least $1 - \delta$,*

$$\sum_{i \in N_{noisy}} \mathbb{1}(F(x_i) \neq y_i) \leq exp\left(-2 \sum_{t=1}^{T} \gamma_t^2\right) + \epsilon(\delta, N, \rho_+, \rho_-).$$

Though our above results are proved under the symmetric error setting, we experimentally verified the performance of our error-resistant boosting procedure.

## 4 A SELF LABELLING FRAMEWORK FOR BOOSTING

We now introduce our Self Labelling framework for boosting. The framework can be broken into two major components: the noisy label generation and the noisy Adaboost algorithm. Section 3 already described the formation and proof of the noisy Adaboost algorithm. In the next two sections go into detail over i) the inference method to generate noisy label and ii) the procedure to estimating the noise levels within the generated labels as inputs into the Adaboost algorithm. More specifically, Section 4.1 delves deeper into detailing the processes of noisy label generation particularly addressesing matrix $\mathcal{L}$ and the aggregation inside of $\mathcal{L}$ that forms the noisy dataset. While Section 4.2 explores how the noise levels within the dataset ($\rho_-, \rho_+$) is calculated and used as inputs to the noisy Adaboost algorithm.

---

**Algorithm 1** Boosting via Self Labelling

---

1: **Input**: Labelled Data $\mathcal{N}_L$, unlabelled Data $\mathcal{N}_U$
2: **Output**: Final Hypothesis $F(x)$
3: For $i = 1, \cdots, M$ :
   - Train $f_i$ on $(x_i, y_i) \in \mathcal{N}_L$
   - Get hypothesis $h_t = f_i(x_i) \rightarrow \{-1, +1\}_{|N_U|}, \ \forall x_i \in \mathcal{N}_U$
   - Add $h_t$ to matrix $\mathcal{L}$
4: Run inference method to generate noisy labels for $\mathcal{N}_U$: $I(\mathcal{L}) \rightarrow \{-1, +1\}_{|N_U|}$. Denote the noisy dataset as $\mathcal{N}_{U_{noisy}}$. Let $\mathcal{N}_{noisy} = \mathcal{N}_L \cup \mathcal{N}_{U_{noisy}}$.
5: Estimate $(\rho_+, \rho_-)$ using Eqn. (6, 7).
6: Initialize: $D_i(1) = 1/N$ for $i \in N_{noisy}$
7: for $t = 1, \cdots, T$:
   - Train weak classifier $g_t$ on data distribution $D(t)$
   - Get weak hypothesis $h_t := g_t(x, i), \ (x_i, \tilde{y}_i) \in \mathcal{N}_{noisy} \rightarrow \{-1, +1\}$
   - Get weighted error $\epsilon^+, \epsilon^-$ using Eqn.(3)
   - Calculate $\omega_+, \omega_-$ and $\alpha_t^+, \alpha_t^-$ (Eqn.(4)), using estimated $\tilde{\rho}_+, \tilde{\rho}_-$.
   - Compute $\alpha_t := \frac{\alpha_t^+ + \alpha_t^-}{2}$.
   - For $i \in N_U$ : Update weight $D_i(t + 1)$ according to Eqn.(5). Else for $i \in N_L$, update weight $D_i(t + 1)$ according to Eqn.(5) with $\rho_+ = \rho_- = 0$.
8: Final Hypothesis:
$$F(x) = sign\left(\frac{\sum_{t=1}^{T} \alpha_t f_t(x)}{T} > 0\right)$$

---

## 4.1 SELF LABELLING

We generate noisy labels for the unlabelled dataset via a crowdsourcing perspective based on the works of (Dawid & Skene, 1979; Liu et al., 2012). On line 3 in Algorithm 1, $\mathcal{M}$ classifiers $\{f_1(\cdot), f_2(\cdot), ...., f_M(\cdot)\}$ are trained on labelled data $\mathcal{N}_L = \{x_1, x_2, \ldots, x_{|N_L|}\}$, with the goal to classify all $\mathcal{N}_U$ data points, $\{x_1, x_2, \ldots, x_{|N_U|}\}$ as closely as possible to their true unknown binary labels $\{y_1, y_2, \ldots, y_{|N_U|}\} \in \{-1, +1\}$. Each data point $x_i \in \mathcal{N}_U$ will receive a noisy label $\tilde{y}_{i,j}$ which denotes classifier $f_j(\cdot) \in M$ prediction on $x_i$.

Let each row in matrix $\mathcal{L} \in \{-1, +1\}^{|N_U| \times M}$, in the final step on line 3, represent $\{\tilde{y}_i\}; \forall i \in N_U$ for each $f_j(\cdot)$ s.t. $\mathcal{L}_{i,j} = f_j(x_i) = \tilde{y}_{i,j}$. Assuming $f_j(\cdot) \neq f_k(\cdot); \forall j, k \in M$ s.t. $j \neq k$ each $f_j(\cdot)$ will learn a different distribution. This will allow for a variation that gives a better prediction on the underlying structure of $N_U$ allowing for some $f_j(\cdot)$ to be closer to capturing the true underlying distribution of $N_U$ compared to others. Assigning weight $q_j$ to each $f_j(\cdot)$ according to its perceived accuracy in respect to other classifiers allows for a more accurate aggregation of all $f_j(\cdot)$ responses. More formally, in step 4 in Algorithm 1, for each of the $\mathcal{M}$ classifiers $\{f_1(\cdot), f_2(\cdot), \ldots, f_M(\cdot)\}$, $I$ (variational inference method) assigns an optimal weight $\{q_1, q_2, \ldots, q_M\}$ such that taking the aggregation of each classifiers prediction for point $x_i \in \mathcal{N}_U$ against its respective weight $q_j$ will result in a value $\tilde{y}_i$ that will be as close as possible to the unknown true label $y_i$. The set of $x_i$ and aggregated $\tilde{y}_i \ \forall i \in \mathcal{N}_U$ will form $\mathcal{N}_{U_{noisy}}$.

Conceptually, $I$ uses $q_j$ to classify $f_j(\cdot)$ as an expert if $q_j > 0.5$, a spammer if $q_j \approx 0.5$ or an adversary if $q_j < 0.5$. If $f_j(\cdot)$ is designated a spammer then $f_j(\cdot)$ is randomly guessing and does not provide any useful prediction. If $f_j(\cdot)$ is denoted as an adversary, $f_j(\cdot)$ is believed to be "purposely" picking the incorrect label and $\tilde{y}_{i,j} = f_j(x_i)$. As the number of labelled data gets smaller, the number of spammers and adversaries found in $M$ classifiers increases.

Assuming conditional independence among classifiers $I$ can predict the optimal weight as following: If $\mathbb{P}(f_j(x), f_k(x)|y) = \mathbb{P}(f_j(x)|y)\mathbb{P}(f_k(x)|y)$ where $j, k \in |\mathcal{M}| s.t. j \neq k$ (treating each classifier as an independent labeler), we can apply inference approaches to aggregate and infer the true label:

$$\hat{q} = \arg\max \log \mathbb{P}(q|\mathcal{L}, \theta) = \log \sum_y \mathbb{P}(q, y|\mathcal{L}, \theta)$$

Expectation maximization can be used to solve for this maximum a posteriori estimator $\hat{q}$ by treating the true labels $y_i$ as the hidden variable. Assuming a Beta($\sigma, \beta$) distribution, an EM can be formulated as follows:

$$\textbf{E Step: } \mu_i(y_i) \propto \prod_{j \in M_i} \hat{q}_j^{\delta_{i,j}}(1 - \hat{q}_j)^{1-\delta_{i,j}} \quad \textbf{M Step: } \hat{q}_j = \frac{\sum_{i \in N_j} \mu_i(\mathcal{L}_{i,j}) + \sigma - 1}{|N_j| + \sigma + \beta - 2}$$

where $\delta_{i,j} = \mathbb{I}[\mathcal{L}_{i,j} = y_i]$ and $\tilde{y}_i$ is estimated by $\tilde{y}_i = \arg\max_{y_i} \mu_i(y_i)$ given $\mu_i$ is the estimated likelihood of $y_i$. $N_j$ stands for all labels observed by classifier $j$: $N_j = (\{f_j(x_i)\}, \forall i \in 1 \cdots |N_U|)$.

Finally, step 5 of Algorithm 1 shows the noisy dataset $(x_i, \tilde{y}_i), \forall i \in \mathcal{N}_{noisy}$ where $\tilde{y}_i$ is the noisy label provided by the inference algorithm if $x_i \in \mathcal{N}_U$, else $\tilde{y}_i = y_i$ if $(x_i, y_i) \in \mathcal{N}_L$. $\mathcal{N}_{noisy}$ then feeds into the Noisy Adaboost Algorithm introduced in Section 3.

### 4.2 Error estimation

Let $\mathcal{A}(.)$ represent the noise resistant AdaBoost algorithm we introduced in section 3. Let $I$ represent the inference method we will use to aggregate matrix $\mathcal{L}$. Inference method $I$ outputs a noisy labelled data set: $N_{U_{noisy}} = \{x_i, \tilde{y}_i\}, \forall x_i \in \mathcal{N}_U$. In order to run $\mathcal{A}$ we need to accurately estimate the error rates $\rho_-, \rho_+$: $\rho_- = (\tilde{y}_i = +1 | y_i = -1)$, $\rho_+ = (\tilde{y}_i = -1 | y_i = +1)$ within $\mathcal{N}_{U_{noisy}}$. Assuming homogeneous error rates, we can derive an accurate estimation for the error rates of $\mathcal{N}_{U_{noisy}}$ from the confusion matrices of the the classifiers $f_j \in \mathcal{M}$.

Denote the false positive and false negative of $f_j(\cdot)$ by $\rho_{+,f_j}$ and $\rho_{-,f_j}$ respectively. The inference algorithm $I$ will output $\rho_{-,f_j}$ and $\rho_{+,f_j}$ for each $f_i \in \mathcal{M}$ by comparing $\{x_i, f_j(x_i)\}$ against the aggregated label from the inference algorithm. $\mathbb{P}(y_i = -1)$ and $\mathbb{P}(y_i = +1)$ are the marginal distribution of positive and negative labels within the dataset. If $\mathbb{P}(y_i = -1)$ and $\mathbb{P}(y_i = +1)$ are known we show the below equation uniquely identify the error rates given any $f_j \in \mathcal{M}$:

**Lemma 1** $\rho_+, \rho_-$ *can be determined by the following set of equations:*

$$\rho_+ = \frac{(\rho_{-,f_j}\mathbb{P}(\tilde{y}_i = 1) - \mathbb{P}(y_{i,f_j} = \tilde{y}_i = 1)}{\mathbb{P}(y_i = +1)(1 - \rho_{+,f_j}) - \mathbb{P}(y_i = -1)(\rho_{-,f_j})} \tag{6}$$

$$\rho_- = \frac{-\mathbb{P}(\tilde{y}_i = 1)(1 - \rho_{+,f_j}) + \mathbb{P}(y_{i,f_j} = \tilde{y}_i = 1)}{-\mathbb{P}(y_i = +1)(1 - \rho_{+,f_j}) + \mathbb{P}(y_i = -1)(\rho_{-,f_j})} \tag{7}$$

In practice we can balance the dataset to make $\mathbb{P}(y_i = -1) = \mathbb{P}(y_i = +1) = 0.5$. The probability terms $\mathbb{P}(\tilde{y}_i = 1)$ and $\mathbb{P}(y_{i,f_j} = \tilde{y}_i = 1)n$ can be estimated through the data. The estimated parameters are plugged into Eqn.(6,7) to approximate $\tilde{\rho}_+, \tilde{\rho}_-$. Note this estimation is done without using ground truth labels.

## 5 Experimental results

The focus of BSL is to provide a framework which can generate an accurate classifier given very little labelled data. In this section, we conduct extensive experiments to verify the effectiveness of BSL and compare with benchmark algorithms.

### 5.1 Datasets

8 UCI datasets were used to evaluate Boosting via Self-Labelling (BSL). Since Boosting via Self-Labelling works on binary classification problems, we chose datasets which only contained two class labels or turned linear regression datasets into binary labels. The first column of Table 1 has the name of the dataset used and the percentage of data labelled. The Cancer dataset had 567 samples and 30 features. The Diabetes dataset had 768 samples and 8 features. The Thyroid dataset had 215 samples and 5 features. The Heart dataset had 303 samples and 13 features. The German dataset has 1000 samples and 20 features. The Image dataset has 2310 samples and 19 features. The Housing dataset has 506 samples and 13 features. And the Sonar dataset has 208 samples and 60 features.

## 5.2 Experimental Setup

The goal of our experiments was to show the performance improvement we achieved by using the BSL compared to other semi-supervised algorithms. We use classification error rate (measuring the fraction of mis-classified sample points) each model faced as the evaluation measure. Table 1 reports the mean of 20 different runs of the experiment. To measure the performance of each trial, we split the data into 40% test and 60% train. We then broke up the training into increasing percentages of unlabelled/labelled data points. Table with all results can be found in Appendix.

The pre-processing step started by taking out any points that had missing features or missing labels in the entire dataset. As a result, the dataset was completely labelled and each sample had all of its features. Before creating the unlabelled dataset, we split the data into testing and training set. The unlabelled data was created by removing the labels from a designated percentage of the training set. Each of the features was normalized between 0 and 1 to allow for a better approximation of the data. At the end of the pre-processing step, three separate lists were outputted: the testing set, the set that contained the unlabelled data points and the set that contained the labelled data points.

During the first step of our framework, the labelled data was passed to a set of supervised machine learning algorithms. Although the number of classifiers could have been theoretically infinite, we chose to limit the number to 10. As a result, the experiment was able to be conducted in a reasonable time. The supervised algorithms were all implemented using the Scikit-learn (Pedregosa et al., 2011) and consisted of KNN, Decision Tree, Gaussian Mixture Model, Naive Bayes, SVM and Logistic Regression. Some of the models were repeated in their use but took on different initialization values to create different classifiers.

The second step of the framework called a basic inference algorithm abbreviated (D&S) (Dawid & Skene, 1979). The crowdsourcing algorithm outputted the noisy labels for each of the unlabelled data points.

The final part of the framework, our noise-resistant variant of the AdaBoost algorithm is used. The alpha value for each classifier was optimised to compensate for the increase in values from the loss functions. We limited the algorithm to running only 20 decision stumps as the base classifier.

Table 1 shows the performance of BSL compared to other state of the art algorithms which try to create optimised classifiers within the label limited dataset. Table 1 only reports the classification errors for 10% of the training data labelled. Each algorithm bench marked a different part of the framework. C-SVM(Liu et al.) showed the improvement noise resistant version of AdaBoost gave over other noise-resistant algorithms. Semiboost, S3VM, Label Propagation, NN, and Logitboost tested against the final classifier produced by the framework. CSVM, Semiboost, S3VM, and Logitboost were all implemented using different external libraries. Label Propagation was implemented using Scikit-Learn (Pedregosa et al., 2011) And the Supervised Neural Network (NN) was was implemented with Keras (Chollet et al., 2015).

## 5.3 Results

**Performance Comparison of BSL with Benchmark Algorithms**    We compared BSL's performance to seven benchmark algorithms, specifically: DS+AdaBoost, Semiboost, S3VM, CSVM, Label Propagation, NN, Logitboost. DS + AdaBoost applied inference method to assign noisy labels to the unlabelled data and used a standard AdaBoost algorithm (Pedregosa et al., 2011) to create a final classifier. The Supervised Neural Network (NN) is a sequential model with two hidden layers each with 100 nodes. If DS was not specified then the model performed without any noise labelled data outputted by the crowdsourcing algorithm. BSL's improvement in performance compared to the other algorithms is significant. BSL consistently had a $\mathbf{20\% - 30\%}$ increase in performance compared to most of the competing algorithms. While BSL is able to outperform most of the algorithms on each of the 8 datasets, it was outperformed by the DS + AdaBoost for trials in the thyroid and sonar dataset. Semiboost was also able to produce better results against BSL in the image dataset. It is important to note that although DS+AdaBoost and Semiboost outperform BSL, the difference in improvement was not significant. This shows that the loss function that we introduced to the noise resistant algorithm might overfit on the data by taking out more noise than was necessary. The loss of performance could also indicate that the error rates passed into the algorithm were not always close to the actual noise in the dataset or were too high to be effective. Table 2 (located in the

Appendix) shows the estimated error rate the inference crowdsourcing method outputted for each experiment. It is important to notice the noise that exists within the dataset after the crowdsourcing step. Our handling of this noise allows our algorithm to perform on average better than the other benchmark algorithms. When noise was close to 50%, the noise resistant Adaboost loss function becomes unbounded and this creates unstable classifications. Similarly, the low error rates present within the thyroid dataset allowed for a non-noise resistant version of Adaboost to outperform BSL.

**Performance with increment in the percentage of unlabelled data**    With Figure 1 (located in the Appendix), we also show the performance of BSL on 4 UCI datasets we used to compare framework against the baseline algorithms with increasing amount of unlabelled data instances. In the experiment, we increase the percentage of unlabelled data one at each step, starting from $50\%$ going up to $99\%$ (or as far as possible). Under each step of the experiment, we ran BSL 20 times randomising at every turn which points were used in training, testing, being labelled or unlabelled. The solid line shows the mean of running the algorithm 20 times at each step. The graph shows that despite the decrease of labelled data available to train on, the framework can maintain relatively similar classification error rate. This feature is significant because it shows that having a lot of unlabelled data does not restrict the performance of the framework and therefore is not a prerequisite for it to perform well. One can note that all the graphs don't go to 0.99% unlabelled data. Since the UCI datasets were not significantly big, the closer we are to 0.99% our training data became single-classed labelled. As a result, our framework could not create a final classifier as the ensemble we used was not able to train on the single-classed dataset. As a result, we stopped the experiment at the points where we started getting a lot of single-class labelled data warnings.

| % Labelled | D&S+AdaB | BSL | D&S+CSVM | Semiboost | S3VM | LP | NN | Logitboost |
|---|---|---|---|---|---|---|---|---|
| Cancer - 10% | 39.1 | **10.22** | 38.16 | 29.65 | 37.63 | 32.68 | 35.94 | 34.65 |
| Diabetes - 10% | 30.19 | **27.79** | 34.98 | 32.97 | 33.9 | 35.24 | 35.63 | 31.90 |
| Thyroid - 10% | **14.51** | 23.94 | 31.32 | 30.83 | 27.21 | 27.91 | 35.18 | 24.25 |
| Heart - 10% | 32.82 | **23.21** | 45.05 | 35.25 | 47.05 | 29.47 | 35.38 | 29.81 |
| German - 10% | 27.5 | **26.75** | 32.25 | 32.25 | 32.25 | 32.49 | 35.82 | 30.28 |
| Image - 10% | 27.99 | 27.06 | 26.17 | **25.31** | 29.74 | 42.98 | 41.71 | 37.28 |
| Housing - 10% | 29.00 | **24.35** | 27.31 | 26.08 | 26.50 | 34.74 | 42.14 | 27.13 |
| Sonar - 10% | **25.60** | 26.24 | 40.42 | 40.53 | 38.19 | 44.68 | 48.81 | 45.12 |

Table 1: Classification Error Rate of BSL and the seven benchmark algorithms. The first column shows the dataset and percent of data labelled. The columns D&S+CSVM, Semiboost, S3VM and LP (label propagation), Logitboost, NN show the performance of the six benchmark algorithms. Each entry shows the mean classification error rate over 20 trials. Full table available in the Appendix.

## 6    DISCUSSION AND CONCLUDING REMARKS

Our goal in this paper is to present a novel and efficient boosting algorithm for semi-labelled datasets and show its effectiveness in providing an accurate classifier. The usefulness of BSL stems in its ability to produce high-quality noisy labels to unlabelled instances and its ability to handle the noises in labels, despite having severely limited amount of labelled data.

Our results over the 8 UCI datasets reveal the performance improvement BSL brings compared to other algorithms currently in use. Our experiments also show how impervious BSL can be to noise by showing constant performance despite increases in unlablled data. A natural direction would be considering datasets that are fully unlabelled. Instead of having an ensemble of supervised classifiers BSL could consider using a consensus clustering approach and view each clustering algorithm as a potential agent within the crowdsource setting. Another important extension of this project would be using a more proficient inference method to extract the labels less nosier than those produced during our experiments. It is also a very interesting question to further study the theoretical guarantees of BSL in more sophisticated settings. Finally, another aspect of the paper we wish to further pursue is looking at non-homogeneous error rates.

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

# A  PROOF FOR THEOREM 1

For our error-corrected AdaBoost we first prove

$$Z := \prod_{t=1}^{T} Z_t = \sum_{i \in N_{noisy}} \exp\big(-\tilde{\ell}(F(x_i), \tilde{y}_i)\big) \tag{8}$$

Following standard argument of AdaBoost:

$$D_i(t+1) = \frac{D_i(t)\exp\big(-\alpha_t \tilde{\ell}(f_t(x_i), \tilde{y}_i)\big)}{Z_t}$$

$$= \frac{D_i(t-1)\exp\big(-\alpha_t \tilde{\ell}(f_t(x_i), \tilde{y}_i) - \alpha_{t-1}\tilde{\ell}(f_{t-1}(x_i), \tilde{y}_i)\big)}{Z_t Z_{t-1}}$$

$$= ...$$

$$= \frac{D_i(1)\exp\big(-\sum_\tau \alpha_\tau \tilde{\ell}(f_\tau(x_i), \tilde{y}_i)\big)}{Z}$$

Since $\tilde{\ell}(f_t(x_i), \tilde{y}_i)$ is linear in $f_t$ we know

$$\sum_{\tau=1}^{t} -\alpha_t \tilde{\ell}(f_\tau(x_i), \tilde{y}_i) = -\tilde{\ell}\big(\sum_{\tau=1}^{t} \alpha_\tau f_\tau(x_i), \tilde{y}_i\big) = -\tilde{\ell}\big(F(x_i), \tilde{y}_i\big)$$

Therefore

$$1 = \sum_{i \in N_{noisy}} D_i(t+1) = \frac{\sum_{i \in N_{noisy}} D_i(1) \cdot \exp\big(-\sum_{\tau=1}^{t} \tilde{\ell}(F(x_i), \tilde{y}_i)\big)}{Z}$$

Multiple $Z$ on both sides, we have proved Eqn. (8).

Define

$$\hat{\ell}\big(f(x_i), \tilde{y}_i = +1\big) := \frac{(1-\rho_-)\mathbb{1}(f(x_i) \neq +1) - \rho_+ \mathbb{1}(f(x_i) \neq -1)}{1 - \rho_+ - \rho_-}, \tag{9}$$

$$\hat{\ell}\big(f(x_i), \tilde{y}_i = -1\big) := \frac{(1-\rho_+)\mathbb{1}(f(x_i) \neq -1) - \rho_- \mathbb{1}(f(x_i) \neq +1)}{1 - \rho_+ - \rho_-}. \tag{10}$$

Next we show that

$$\sum_{i \in N_{noisy}} \exp\big(-\tilde{\ell}(F(x_i), \tilde{y}_i)\big) \geq \sum_{i \in N_{noisy}} \hat{\ell}\big(F(x_i), \tilde{y}_i\big)$$

When $F(x_i) = \tilde{y}_i = +1$, $\hat{\ell}(F(x_i), \tilde{y}_i) = 1 - \omega_1 < 0$, but $\exp\big(-\tilde{\ell}(F(x_i), \tilde{y}_i)\big) > 0$. When $F(x_i) = -1, \tilde{y}_i = +1$, $\hat{\ell}(F(x_i), \tilde{y}_i) = \omega_1$, but $\exp\big(-\tilde{\ell}(F(x_i), \tilde{y}_i)\big) > \exp(\omega_1) > \omega_1$. The case for $\tilde{y}_i = 1$ is symmetric.

Via Hoeffding inequality we know that with high probability at least $1 - \delta$

$$\sum_{i \in N_{noisy}} \hat{\ell}\big(F(x_i), \tilde{y}_i\big) \geq \sum_{i \in N_{noisy}} \mathbb{1}(f(x_i) \neq y_i) - \epsilon(\delta, N, \rho_-, \rho_+)$$

where $\epsilon(\delta, N, \rho_+, \rho_-) := \max(\omega_+, \omega_-) \cdot \sqrt{\frac{N \ln \frac{2}{\delta}}{2}}$.

Therefore

$$\sum_{i \in N_{noisy}} \mathbb{1}(f(x_i) \neq y_i) \tag{11}$$

$$\leq \sum_{i \in N_{noisy}} \hat{\ell}\big(F(x_i), \tilde{y}_i\big) + \epsilon(\delta, N, \rho_-, \rho_+) \tag{12}$$

$$\leq \sum_{i \in N_{noisy}} \exp\big(-\tilde{\ell}(F(x_i), \tilde{y}_i)\big) + \epsilon(\delta, N, \rho_-, \rho_+) \tag{13}$$

$$= Z + \epsilon(\delta, N, \rho_-, \rho_+) \tag{14}$$

Now we prove that

$$Z \leq \exp\left(-2\sum_{t=1}^{T}\gamma_t^2\right)$$

First we notice the following: at time $t$

$$\tilde{\ell}(f(x_i) = +1, \tilde{y}_i = +1) = \exp(-\omega_+\alpha_t) \tag{15}$$
$$\tilde{\ell}(f(x_i) = -1, \tilde{y}_i = +1) = \exp(\omega_+\alpha_t) \tag{16}$$
$$\tilde{\ell}(f(x_i) = -1, \tilde{y}_i = -1) = \exp(-\omega_-\alpha_t) \tag{17}$$
$$\tilde{\ell}(f(x_i) = +1, \tilde{y}_i = -1) = \exp(\omega_-\alpha_t) \tag{18}$$

Then when $\tilde{y}_i = +1$, taking derivatives of $Z_t$ w.r.t. $\alpha_t$

$$\frac{\partial Z_t}{\partial \alpha_t} = \sum_{x \in A_+} -\omega_+\exp(-\omega_+\alpha_t) + \sum_{x \in \bar{A}_+} \omega_+\exp(\omega_+\alpha_t) \tag{19}$$

Here we have defined four sets:

$$A_+ := \text{the set of correctly classified data when } \tilde{y}_i = +1, \tag{20}$$
$$A_- := \text{the set of correctly classified data when } \tilde{y}_i = -1, \tag{21}$$

and $\bar{A}_+, \bar{A}_-$ are their complement sets. Set derivative in Eqn. (19) to 0 we have

$$\alpha_t = \frac{1}{2\omega_+}\ln\frac{1-\hat{\epsilon}_t}{\hat{\epsilon}_t}$$

where $\hat{\epsilon}_t$ is defined as follows:

$$\hat{\epsilon}_t^+ := \mathbb{P}_{\tilde{y}=+1}(f_t(x) \neq \tilde{y})$$

Similarly when $\tilde{y}_i = -1$

$$\hat{\epsilon}_t^- := \mathbb{P}_{\tilde{y}=-1}(f_t(x) \neq \tilde{y})$$

Define $\hat{\epsilon}_t = \max\{\hat{\epsilon}_t^+, \hat{\epsilon}_t^-\}$ and when $\rho_+ = \rho_-$, we have $\omega_+ = \omega_-$. And consequently, $\alpha_t = \alpha_t^+ = \alpha_t^-$ Next follows standard argument in boosting we are ready to prove

$$Z_t \leq \sqrt{\hat{\epsilon}_t(1-\hat{\epsilon}_t)}$$

Without loss of generality, consider the negative label case $\tilde{y}_i = -1$. Then

$$
\begin{aligned}
Z_t &= \hat{\epsilon}_t^-\sqrt{\frac{1-\hat{\epsilon}_t}{\hat{\epsilon}_t}} + (1-\hat{\epsilon}_t^-)\sqrt{\frac{\hat{\epsilon}_t}{1-\hat{\epsilon}_t}} \\
&\leq \hat{\epsilon}_t\sqrt{\frac{1-\hat{\epsilon}_t}{\hat{\epsilon}_t}} + (1-\hat{\epsilon}_t)\sqrt{\frac{\hat{\epsilon}_t}{1-\hat{\epsilon}_t}} \\
&= \sqrt{\hat{\epsilon}_t(1-\hat{\epsilon}_t)}
\end{aligned}
$$

The inequality is due to the fact $\sqrt{\frac{1-\hat{\epsilon}_t}{\hat{\epsilon}_t}} > \sqrt{\frac{\hat{\epsilon}_t}{1-\hat{\epsilon}_t}}$. Therefore

$$Z_t \leq \sqrt{\hat{\epsilon}_t(1-\hat{\epsilon}_t)} = \sqrt{1-4\gamma_t^2},\ \gamma_t := \frac{1}{2} - \hat{\epsilon}_t$$

Further

$$\sqrt{1-4\gamma_t^2} \leq \exp(-2\gamma_t^2)$$

This completes the proof.

PROOF FOR LEMMA 1

Given $\tilde{y}_{i,f_j} = f_j(x_i) \in \{\pm 1\}, x_i \in \mathcal{N}_U, y_i$ is true labels of $x_i \in \mathcal{N}_U$, and $y_i \in (x_i, y_i) \in \mathcal{N}_{U_{noisy}}$.

$$\mathbb{P}(y_{i,f_j} = \tilde{y}_i = 1)$$
$$= \mathbb{P}(y_{i,f_j} = \tilde{y}_i = 1, y_i = 0) + \mathbb{P}(y_{i,f_j} = \tilde{y}_i = 1, y_i = 1)$$
$$= \mathbb{P}(y_{i,f_j} = \tilde{y}_i = 1 | Y = 0) \cdot \mathbb{P}(y_i = 0)$$
$$\quad + \mathbb{P}(y_{i,f_j} = \tilde{y}_i = 1 | y_i = 1) \cdot \mathbb{P}(y_i = 1)$$
$$= \mathbb{P}(y_{i,f_j} = 1 | y_i = 0) \mathbb{P} \cdot (\tilde{y}_i = 1 | y_i = 0) \cdot \mathbb{P}(y_i = 0)$$
$$\quad + \mathbb{P}(y_{i,f_j} = 1 | y_i = 1) \cdot \mathbb{P}(\tilde{y}_i = 1 | y_i = 1) \cdot \mathbb{P}(y_i = 1)$$
$$= \mathbb{P}(y_i = 0) \cdot \rho_- \cdot \rho_{-,f_j} + \mathbb{P}(y_i = 1) \cdot (1 - \rho_+) \cdot (1 - \rho_{+,f_j})$$

Further we have

$$\mathbb{P}(\tilde{y}_i = 1) = \mathbb{P}(y_i = 0) \cdot \rho_- + \mathbb{P}(y_i = 0)(1 - \rho_+) \tag{22}$$

Solving above linear equations completes the proof.

ADDITIONAL EXPERIMENTAL RESULTS

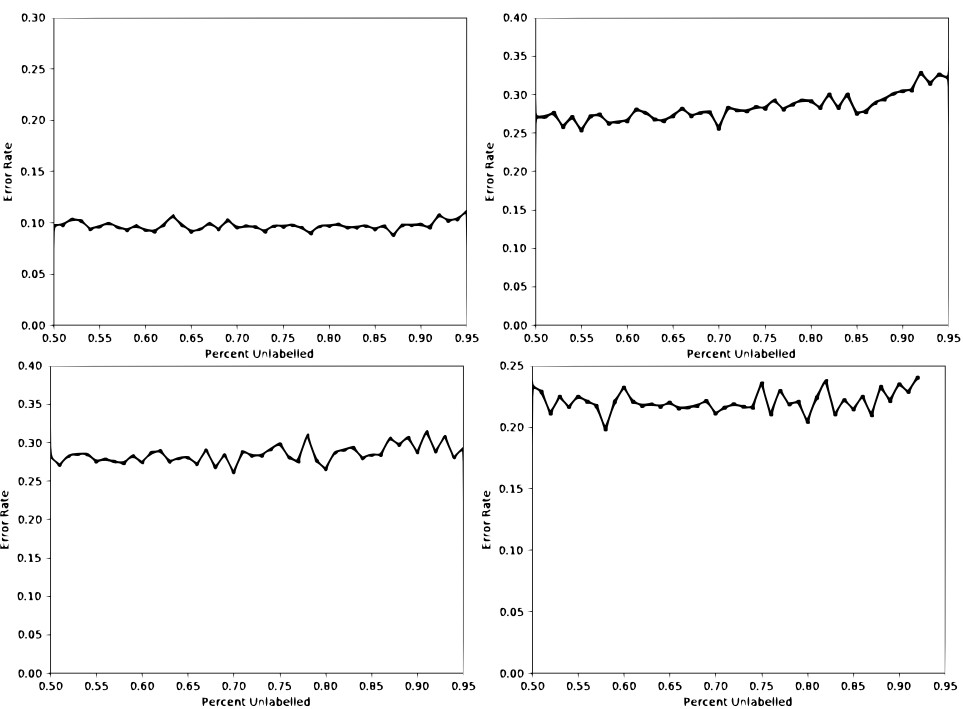

Figure 1: **Top left**: Cancer Dataset, **Top Right**: Diabetes Dataset, **Bottom Left**: Heart Dataset, **Bottom Right**: Thryoid Dataset. All these graphs represent an increasing % of unlabelled data within the dataset.

| % Unlabelled | Estimated Noise Rate | Actual Noise Rate |
|---|---|---|
| Cancer - 10% | 5.57 | 9.68 |
| Cancer - 20% | 3.51 | 5.79 |
| Cancer - 30% | 3.22 | 4.17 |
| Cancer - 40% | 2.93 | 3.64 |
| Diabetes - 10% | 27.39 | 30.38 |
| Diabetes - 20% | 23.47 | 25.88 |
| Diabetes - 30% | 20.00 | 19.03 |
| Diabetes - 40% | 22.39 | 27.46 |
| Thyroid - 10% | 8.52 | 6.15 |
| Thyroid - 20% | 7.75 | 9.55 |
| Thyroid - 30% | 4.65 | 6.75 |
| Thyroid - 40% | 4.51 | 8.63 |
| Heart - 10% | 17.67 | 16.03 |
| Heart - 20% | 14.91 | 13.86 |
| Heart - 30% | 15.81 | 17.20 |
| Heart - 40% | 14.91 | 18.52 |
| German - 10% | 24.83 | 25.81 |
| German - 20% | 24.16 | 21.56 |
| German - 30% | 18.16 | 20.84 |
| German - 40% | 15.83 | 16.09 |
| Image - 10% | 66.01 | 57.57 |
| Image - 20% | 57.50 | 65.23 |
| Image - 30% | 50.43 | 53.72 |
| Image - 40% | 42.71 | 41.04 |
| Housing - 10% | 25.74 | 26.92 |
| Housing - 20% | 14.15 | 9.57 |
| Housing - 30% | 5.61 | 8.81 |
| Housing - 40% | 4.62 | 8.82 |
| Sonar - 10% | 23.78 | 28.26 |
| Sonar - 20% | 26.61 | 29.69 |
| Sonar - 30% | 20.53 | 19.35 |
| Sonar - 40% | 15.32 | 10.62 |

Table 2: Estimated noise within the dataset and the actual noise found within the dataset

| % Labelled | D&S+AdaB | BSL | D&S+CSVM | Semiboost | S3VM | LP | NN | Logitboost |
|---|---|---|---|---|---|---|---|---|
| Cancer - 10% | 39.1 | **10.22** | 38.16 | 29.65 | 37.63 | 32.68 | 35.94 | 34.65 |
| Cancer - 20% | 33.42 | **8.9** | 37.81 | 19.65 | 37.81 | 29.08 | 31.62 | 31.82 |
| Cancer - 30% | 36.73 | **9.34** | 37.02 | 10.92 | 38.03 | 21.56 | 30.97 | 26.53 |
| Cancer - 40% | 35.42 | **9.17** | 39.25 | 10.75 | 36.84 | 18.68 | 28.03 | 17.81 |
| Diabetes - 10% | 30.19 | **27.79** | 34.98 | 32.97 | 33.9 | 35.24 | 35.63 | 31.90 |
| Diabetes - 20% | 28.18 | **26.69** | 35.63 | 33.18 | 32.92 | 32.92 | 31.91 | 28.29 |
| Diabetes - 30% | 27.27 | **25.32** | 34.09 | 33.86 | 35.32 | 30.71 | 31.38 | 27.39 |
| Diabetes - 40% | 27.96 | **26.61** | 34.33 | 31.94 | 34.68 | 30.31 | 30.43 | 28.81 |
| Thyroid - 10% | **14.51** | 23.94 | 31.32 | 30.83 | 27.21 | 27.91 | 35.18 | 24.25 |
| Thyroid - 20% | **10.51** | 21.09 | 30.74 | 29.84 | 32.09 | 26.74 | 34.31 | 25.34 |
| Thyroid - 30% | **9.49** | 20.58 | 31.05 | 30.08 | 29.53 | 25.58 | 26.82 | 27.46 |
| Thyroid - 40% | **8.91** | 21.12 | 31.63 | 30 | 30.7 | 22.09 | 23.41 | 29.78 |
| Heart - 10% | 32.82 | **23.21** | 45.05 | 35.25 | 47.05 | 29.47 | 35.38 | 29.81 |
| Heart - 20% | 26.23 | **20.49** | 44.26 | 29.51 | 45.9 | 29.26 | 34.40 | 30.56 |
| Heart - 30% | 25.41 | **20.49** | 45.08 | 27.05 | 45.74 | 27.88 | 23.87 | 36.67 |
| Heart - 40% | 20.49 | **15.57** | 43.86 | 22.95 | 45.9 | 26.31 | 23.82 | 41.32 |
| German - 10% | 27.5 | **26.75** | 32.25 | 32.25 | 32.25 | 32.49 | 35.82 | 30.28 |
| German - 20% | 26.5 | **25.8** | 27.00 | 27.00 | 27.00 | 26.24 | 31.33 | 29.12 |
| German - 30% | 24.25 | **22.47** | 26.54 | 25.42 | 27.18 | 26.34 | 29.17 | 28.55 |
| German - 40% | 24.75 | **20.59** | 25.84 | 25.11 | 25.39 | 26.81 | 28.11 | 27.31 |
| Image - 10% | 27.99 | 27.06 | 26.17 | **25.31** | 29.74 | 42.98 | 41.71 | 37.28 |
| Image - 20% | 27.75 | 26.48 | **24.17** | 24.67 | 28.44 | 37.38 | 38.22 | 36.19 |
| Image - 30% | 25.10 | 25.83 | 23.85 | **22.83** | 25.15 | 35.06 | 31.06 | 33.45 |
| Image - 40% | 23.53 | 24.64 | 24.80 | **21.94** | 22.87 | 32.51 | 27.57 | 29.44 |
| Housing - 10% | 29.00 | **24.35** | 27.31 | 26.08 | 26.50 | 34.74 | 42.14 | 27.13 |
| Housing - 20% | 21.64 | **20.42** | 23.31 | 25.96 | 24.50 | 35.35 | 39.45 | 25.62 |
| Housing - 30% | 21.64 | **19.64** | 23.41 | 25.53 | 23.41 | 36.79 | 31.87 | 25.29 |
| Housing - 40% | 20.16 | **19.16** | 22.09 | 24.34 | 21.04 | 33.89 | 26.71 | 23.48 |
| Sonar - 10% | **25.60** | 26.24 | 40.42 | 40.53 | 38.19 | 44.68 | 48.81 | 45.12 |
| Sonar - 20% | **16.60** | 18.39 | 36.87 | 33.94 | 37.73 | 38.07 | 31.72 | 31.69 |
| Sonar - 30% | **13.34** | 14.1 | 29.00 | 27.87 | 30.32 | 33.33 | 25.83 | 25.88 |
| Sonar - 40% | **10.63** | 13.43 | 26.17 | 25.78 | 26.17 | 26.24 | 21.64 | 24.68 |

Table 3: Classification Error Rate of BSL and the seven benchmark algorithms. The first column shows the dataset and percent of data labelled. The columns D&S+CSVM, Semiboost, S3VM and LP (label propagation), Logitboost, NN show the performance of the six benchmark algorithms. Each entry shows the mean Classifcation error rate over 20 trials.

