# OpenReview forum: "Semi-Supervised Boosting via Self Labelling"
_ICLR.cc/2020/Conference — Reject_

### Official Review · AnonReviewer1 · 2019-10-22
**Official Blind Review #1**

**Rating:** 1

**Review:**

In this paper, the authors present an approach for semi-supervised learning which combines noisy labels with boosting. In a first step, the labeled instances are used to train a set of classifiers, and these are used to create noisy labels for the unlabeled instances. Then, an EM procedure is used to estimate the noise level of each instance. Finally, a version of AdaBoost which accounts for instance noise levels is proposed to create a final classifier. A limited set of experiments suggests the proposed approach is competitive with existing approaches.

Major Comments

As a non-expert in this area, I had trouble identifying the novel contributions of this work. For example, many of the results in Section 3 (noise-resistant AdaBoost) seem to replicate, or follow closely, the results of [Natarajan et al., 2013]. Similarly, using EM to assign pseudo-labels has been extensively studied in the literature [Lee, WREPL 2013; Chapelle and Zien, AISTATS 2005; Kang et al., ECCV 2018; Rottman et al., ICMLA 2018].


The experiments are very poorly described, so it is difficult to gauge if they are valid:

Most importantly, the authors point out that the estimated error rates do not always match the actual error rates. Since this seems to be one of the most important factors of the proposed approach, further investigation should be performed to answer questions like: why is the error rate not estimated well? on what type of datasets? can more/better supervised learners help? In some cases (Diabetes, Thyroid, Heart), the actual noise rate increases with more labeled samples. What does that mean?

Second, the proposed approach seems to have a number of important hyperparameters, including the number of supervised models trained and their hyperparameters, the parameters of the Beta distribution used as a prior on the noise estimation, and the hyperparameters of the AdaBoost algorithm. Likewise, all of the competing algorithms also have hyperparameters which are known to affect performance (e.g., learning rate for NNs). The paper does not mention how (or if) a validation set was used to select these.

Third, while the caption of Table 1 mentions that 20 trials were used, it is not clear if this was some sort of k-fold cross validation, Monte Carlo, cross validation, the same splits but with different random seeds, etc. Additionally, the variance across the different trials should be given; otherwise, it is not possible to tell if any of the empirical results are significant.

Minor Comments:

The references are not consistently formatted.

This paper is very notation heavy. It would be helpful to include a “table of symbols” for the reader in an appendix.

Additionally, the notation in the paper is not consistent. For example, both “$M$” and “$\mathcal{M}$ are used to indicate the number of models trained on the labeled data. Later on, “$\mathcal{M}$” is also used to refer to the set of trained classifiers. The first bullet point in Step 3 of the pseudocode seems to suggest that each classifier is trained on a single labeled data point. The equation at the bottom of Page 5 used \theta, but it does not seem to be defined.

The discussion on experts, spammers, and adversaries could be helpful if this terminology were used throughout the paper; however, it is used in only one paragraph.

The main body of the paper should mention that proofs are given in the appendices.

For context, if may be helpful to mention that graph convolutional networks and other representation learning techniques are commonly used for semi-supervised learning (e.g., [Kipf and Welling, ICML 2016]). Those approaches are quite different (and lack any sort of theoretical guarantees, for the most part), though, so empirical comparisons may not be so meaningful.

It would be helpful to give a sentence or two on the intuition behind what the proofs are showing. For a non-expert, they are very difficult to follow.

Do the various proofs still hold when the datasets are artificially balanced (with respect to the last paragraph in Section 4)?

It would be helpful to include the performance using the complete labeled dataset for comparison.

Stratified sampling could be used to ensure both classes are present in the training data. Also, “0.99%” -> “99%”.

Besides accuracy, some measure like AuROC or the F1 score which account for class imbalance should be given.

Typos, etc.

“Logitboost tested against” -> “Logitboost were tested against”


“therefore is not” -> “therefore, having a lot of labeled data is not”




**Experience Assessment:**

I do not know much about this area.

**Review Assessment: Checking Correctness Of Derivations And Theory:**

I did not assess the derivations or theory.

**Review Assessment: Checking Correctness Of Experiments:**

I carefully checked the experiments.

**Review Assessment: Thoroughness In Paper Reading:**

I read the paper at least twice and used my best judgement in assessing the paper.

---

> ### Author Response · Authors · 2019-11-09
> **Addressing Some Concerns Over Error Rate**
>
> The parameters on the model could have been explained better. The 20 experiments was running the experiment 20 times with a random split occurring every time. Variance was something that could have helped add, but is not extremely important.
>
> The point of the estimated error and actual error was that they were very close. You cannot get an exact estimate.

---

> > ### Comment · AnonReviewer1 · 2019-11-15
> > **RE: Author response**
> >
> > I have read the other reviews and authors' responses. They do not change my view of the paper.

---

### Official Review · AnonReviewer3 · 2019-10-23
**Official Blind Review #3**

**Rating:** 1

**Review:**

The authors propose a new semi-supervised boosting approach. The approach takes a set of supervised learning algorithms to simulate "crowd-source" labels of the unlabeled data, which are then used to generate a noisy label per unlabeled instance. The noise level is then estimated with an agreement-based scheme, and fed to a modified AdaBoost algorithm that is more noise-tolerant given the noise level. Some theoretical guarantee of the modified AdaBoost algorithm is derived and promising experiment results are demonstrated.

My suggestion is to reject the paper, with the following key reasons.

(1) Contribution is insufficient, or perhaps not well-highlighted. For the three pieces of contribution, Section 4.1 (self-labeling, which is highlighted within the title) seems to be a trivial borrowing of an existing idea in crowd sourcing from 1979. It is not clear whether Section 4.2 (error estimation) is an original contribution or not, but even if it is original Lemma 1 seems marginally trivial. Section 3 (noise-resistant AdaBoost) plugs a known surrogate loss for noisy labels into AdaBoost. But despite the ugly math, the results seem to be equivalent to a heuristically-shrunk alpha_t for AdaBoost. None of the pieces seem to make a solid contribution to the problem of interest.

(2) Assumptions are not reasonable. Section 3 and Sections 4.2 both rely on "homogeneous error rates" which does not seem to be the case when the noise is generated from classifier-target mismatch. In particular, the noisy will only happen in mismatch areas, and not happen in other areas, making it non-homogeneous. The authors did not discuss the rationality of this assumption and/or how it affects the designed approach. In Section 4.2, there is another assumption that "in practice we can balance the dataset", which might be true for the labeled part through sampling, but not necessarily true for the unlabeled part. So it is not clear whether this assumption can be met. Section 4.2 also assumes that "the probability can be estimated through the data" but did not mention how large the data needs to be for an accurate estimation.

(3) Experiments cannot be easily replicated. To begin with, the authors claim to use 10 classifiers from scikit-learn as the initial labeler, but the exact 10 (including parameters) are not pinged down. In the data sets, there is a procedure "or turned linear regression datasets into binary labels" that does not seem sufficiently clear for replication. It is not clear whether "feature normalization" considers only the training set or the whole training+test set.

Having said that, there are some other suggestions:

(4) Writing needs improvement. Many of the parts contains unnecessarily ugly math notations without motivation. Even the core Section 3 looks like a LaTeX math demo than a clear illustration of scientific ideas.

(5) It is not clear what the importance of Theorem 1 is. There doesn't seem to be a guarantee of gamma_t > 0 given the authors' definition of hat{epsilon}_t (worse case error of the two classes), and then the first part of Theorem 1 is not fast decreasing. It is not clear whether the N in the second term is N_noisy. In any case, the theorem is not clearly described enough to help understand the contribution of the paper.

(6) A baseline that should be considered is to treat the noisy labels as "soft labels" and then apply confidence-based boosting.

Improved Boosting Algorithms Using Confidence-rated Predictions, Schapire and Singer 1999.


**Experience Assessment:**

I have read many papers in this area.

**Review Assessment: Checking Correctness Of Derivations And Theory:**

I assessed the sensibility of the derivations and theory.

**Review Assessment: Checking Correctness Of Experiments:**

I carefully checked the experiments.

**Review Assessment: Thoroughness In Paper Reading:**

I read the paper at least twice and used my best judgement in assessing the paper.

---

> ### Author Response · Authors · 2019-11-09
> **Clarification of How we conducted the Experiments**
>
> Turning the linear regression into binary case was simply putting a limit whether the price was over a certain limit or not.
>
> The parameters for many of the classifiers were based on the original values that were given based on the scikit learn models.

---

> > ### Comment · AnonReviewer3 · 2019-11-09
> > **still not reproducible**
> >
> > Thank the authors for clarifying. But even when taking the clarification into account, I'd still be shocked if any educated researcher can reproduce the experiments given the details in the paper/comments. I definitely suggest the authors to put more focus on reproducibility. Thanks.

---

### Official Review · AnonReviewer2 · 2019-10-23
**Official Blind Review #2**

**Rating:** 3

**Review:**

The paper proposed a method combining boosting with semi-supervised learning to handle classification problems when only partial data points have labels available. The method first trains a classifier with the true labels, and predicts labels for unlabeled data (with some error rate), then a bias boosting method is applied on the larger dataset to construct the final classifier. I find the topic interesting but I'm concerned about the novelty level of the paper. Here some further comments.

1. Theorem 1 seems interesting and it will form a strong result if the assumption \rhp_{+{ = \rho_{-} is removed.

2. Lemma 1 "in practice ...", how to balance the dataset to make sure the two class have similar size? The labeled data can be tailored to ensure this, but one cannot make it happen for the unlabeled data.

3. In the experiments, UCI datasets seem not comprehensive to demonstrate the advantage of the proposed method. More datasets with higher volume could be better.  Also, how is the result compared to the case when all the training data labels are known? What is the gap like?

4. Some typos and writing issues, like equation (6) unbalanced brackets.

**Experience Assessment:**

I have read many papers in this area.

**Review Assessment: Checking Correctness Of Derivations And Theory:**

I assessed the sensibility of the derivations and theory.

**Review Assessment: Checking Correctness Of Experiments:**

I assessed the sensibility of the experiments.

**Review Assessment: Thoroughness In Paper Reading:**

I read the paper at least twice and used my best judgement in assessing the paper.

---

> ### Author Response · Authors · 2019-11-09
> **Clarification of Our Novelty**
>
> The novelty in this paper lies in a couple different avenues. Overall this paper is implemented as a framework to help increase the overall prediction accuracy on semi-supervised data. Beyond the framework, the paper produced a novel approach of applying the Natarajan et al. 2013 loss function to a set of supervised learning algorithms under a crowdsourcing environment. The theoretical proof and implementation of this function are shown to perform within the experimental section compared to other semi-supervised algorithms.
>
> We evaluated our paper on the same datasets that were used in the 2013 paper.
>
> Multiple works that achieve 99 percent accuracy when all training data known are not something that would be interesting to put in the paper as our semi-supervised methods would not compare very well.

---

### Decision · Program_Chairs · 2019-12-19

**Decision:**

Reject

**Comment:**

The paper presents a new semi-supervised boosting approach.

As reviewers pointed out and AC acknowledge, the paper is not ready to publish in various aspects: (a) limited novelty/contribution, (b) reproducibility issue and (c) arguable assumptions.

Hence, I recommend rejection.